# TLR7 controls myeloid-derived suppressor cells expansion and function in the lung of C57BL6 mice infected with *Schistosoma japonicum*

Lu Zhou[1‡], Yiqiang Zhu[1‡], Iengshan Mo[1‡], Mei Wang[1], Jie Lin[1], Yi Zhao[2], Yuanfa Feng[1], Anqi Xie[1], Haixia Wei[1], Huaina Qiu[1], Jun Huang[1,3]*, Quan Yang [1,2,3,4]*

**1** Department of Pathogenic Biology and Immunology, School of Basic Medical Sciences, Guangzhou Medical University, Guangzhou, China, **2** Sino-French Hoffmann Institute, Guangzhou Medical University, Guangzhou, China, **3** The State Key Laboratory of Respiratory Disease, The First Affiliated Hospital, Guangzhou Medical University, Guangzhou, China, **4** Guangzhou Municipal and Guangdong Provincial Key Laboratory of Protein Modification and Degradation, School of Basic Medical Sciences, Guangzhou Medical University, Guangzhou, China

‡ These authors share equal first authorship on this work.
* hj165@sina.com (JH); yquangy2015@gzhmu.edu.cn (QY)

**Data Availability Statement:** All relevant data are within the manuscript and its Supporting Information files.

## Abstract

Toll-like receptors (TLRs) play an important role in the induction of innate and adaptive immune responses against *Schistosoma japonicum* (*S. japonicum*) infection. However, the role of Toll-like receptor 7 (TLR7) in the mouse lung during *S. japonicum* infection and the myeloid-derived suppressor cells (MDSCs) affected by the absence of TLR7 are not clearly understood. In this study, the results indicated that the MDSCs were accumulated and the proportion and activation of CD4+ and CD8+ T cells were decreased in the lung of mice at 6–7 weeks after *S. japonicum* infection. Then, the expression of TLR7 was detected in isolated pulmonary MDSCs and the results showed that the expression of TLR7 in MDSCs was increased after infection. Furthermore, TLR7 agonist R848 could down-regulate the induction effect of the soluble egg antigen (SEA) on pulmonary MDSCs in vitro. Meanwhile, TLR7 deficiency could promote the pulmonary MDSCs expansion and function by up-regulating the expression of PD-L1/2 and secreting of IL-10 in the mice infected with *S. japonicum*. Mechanistic studies revealed that *S. japonicum* infection and the antigen effects are mediated by NF-κB signaling. Moreover, TLR7 deficiency aggravates *S. japonicum* infection-induced damage in the lung, with more inflammatory cells infiltration, interstitial dilatation and granuloma in the tissue. In summary, this study indicated that TLR7 signaling inhibits the accumulation and function of MDSCs in *S. japonicum* infected mouse lung by down-regulating the expression of PD-L1/2 and secreting of IL-10, via NF-κB signaling.

## Author summary

Schistosomiasis is a zoonotic parasitic disease that seriously affects human health. Many adults and children with schistosomiasis develop lung symptoms such as coughing, chest

**Funding:** This work was supported by the following grants to QY: Guangzhou Municipal Science and Technology Project (202102080014), Youth project fund of state key laboratory of respiratory diseases (SKLRD-QN-201921), Students' Innovation Ability Enhancement Plan Project of Guangzhou Medical University (PX-66221466), Discipline Construction Fund of Guangzhou Medical University (JCXKJS2021C08), and Key Discipline of Guangzhou Education Bureau (Basic Medicine) (201851839). The funders had no role in study design, data collection and analysis, decision to publish, or preparation of the manuscript.

**Competing interests:** The authors have declared that no competing interests exist.

pain and bloody sputum. In addition to their respiratory and metabolic functions, the lungs also play a role in the immune system. MDSCs play an important role in Schistosoma infection-induced diseases. TLR7 is an intracellular member of the innate immune receptor. The role of TLR7 on MDSCs mediated immune response in the lung is still unclear. Here, our data showed that the percentage and numbers of MDSCs increased in the lung of infected mice, and the expression of TLR7 in pulmonary MDSCs was increased after infection. When TLR7 gene was knockout, the percentage of pulmonary MDSCs was increased after infection and the expression of PD-L1/2 and IL-10 were increased in *S. japonicum* infection-induced pulmonary MDSCs. Additionally, the effects of TLR7 on pulmonary MDSCs are dependent on the activation of NF-κB p65. Finally, we found TLR7 deficiency aggravates *S. japonicum* infection-induced damage in the lung, with more interstitial dilatation, thickened alveolar cavity and granuloma. In this study, the characteristic of MDSCs in the lung of *S. japonicum* infected C57BL/6 mice was explored, and the role of TLR7 on the progress of MDSCs activation and differentiation was investigated.

## Introduction

Schistosomiasis is a neglected public health problem in many developing countries, with high morbidity and mortality. More than 260 million people live with schistosomiasis, and regular mass treatment is used to prevent the disease [1–3]. After infection, schistosoma larvae migrate to different organs, including the lung. Once reaching the adult stage, the fluke lays eggs, which are deposited in the liver, lung, and intestinal wall, inducing granulomatous inflammation, and progressive fibrosis [2,4]. Soluble egg antigen (SEA) is also a main antigen related to *S. japonicum* infection mediates the immune response [5,6].

Schistosomiasis can cause lung damage, and many adults and children with schistosomiasis develop lung symptoms such as coughing, chest pain, and the occasional bloody sputum [7]. The lung is an important respiratory organ in humans and other animals. It is located in the chest cavity, and has abundant capillaries, which are responsible for the exchange of gas between the body and the outside world. Lung mucosa covers a large area and contains a large number of immune cells under the mucosa, including T helper (Th) cells, natural killer (NK) cells, regulatory T cells (Treg), dendritic cells (DC), myeloid-derived suppressor cells (MDSCs), macrophages, and others, which also play an important role in resisting the invasion of pathogens [8–10]. Interestingly, the lung is a reservoir for haematopoietic progenitors, which produce platelets and other immune cells [11,12]. Many immune cells are reported to participate in the process of *S. japonicum*-induced lung injury in the mice. For example, γδ T cells could adjust the Th2 dominant immune response in the lung of *S. japonicum*-infected mice [13]. The Treg cells upregulated by lung-stage *S. mansoni* infection could alleviate allergic airway inflammation [14]. However, the detailed mechanism of *S. japonicum*-induced lung injury is unclear.

MDSCs are a group of heterogeneous cells with immature characteristics and strong immunosuppressive function, which play an important role in the progression and outcome of various diseases such as tumors and infections [15]. MDSCs can inhibit the functions of a variety of immune cells, including T cells, NK cells, macrophages and dendritic cells and induce the production of regulatory T cells (Tregs) [16]. MDSCs are inflammatory cells that secrete a variety of cytokines, including IL-6, IL-1α, GM-CSF, IL-10 and IFN-γ [17,18]. In addition, some pathological conditions, such as tumors and pathogen infections, can induce the expression of

programmed cell death 1 ligand 1/2 (PD-L1/2) in myeloid cells, including MDSCs, leading to T cell apoptosis [19,20]. MDSCs were characterized by the co-expression of myeloid markers Gr1 and CD11b, and could be further divided into monocytes (M-MDSCs) and polymorphonuclear cells (PMN-MDSCs) according to the expression of Ly6G and Ly6C epitopes. Each group has a different biological function and different mechanism involved in its mediation function [15,16]. In addition to tumors, recent studies have shown that some parasites (such as trypanosoma, nematodes and echinococcus) infection can induce the production of MDSCs, and MDSCs plays an important role in the occurrence and development of these parasitic diseases [21–23]. Our previous study found a large number of MDSCs aggregations in the spleen and lymph nodes of mice infected with *S. japonicum* [24]. However, the role and mechanism of MDSCs in the lung lesions caused by *S. japonicum* is not clear.

Pattern recognition receptors (PRRs), such as Toll-like receptors (TLRs), have been reported to recognize pathogen-associated molecular patterns (PAMPs) of Schistosoma and modulate the immune response [25,26]. TLR7 is an endosomal TLR that recognizes single-stranded RNA (ssRNA) and mediates early innate immune responses to viruses, bacteria and malaria [27,28]. Recently, TLR7 agonists were found to be therapeutics against viral infections and bacteria [29,30]. Moreover, TLR7 agonists reverse oxaliplatin resistance in colorectal cancer by directing the MDSCs to tumorigenic M1-macrophages [31]. In addition, the absence of TLR7 can increase MDSCs accumulation and Th2 biased response to influenza A virus infection in the mice [32]. Moreover, TLR7 modulates B-cell immune responses in the spleen of C57BL/6 mice infected with *S. japonicum* [33].

In this study, the characteristics of MDSCs in the lung of *S. japonicum* infected C57BL/6 mice were explored, and the role of TLR7 on the progression of MDSCs activation and differentiation was investigated.

## Materials and methods

### Ethics statement

All animal work was approved by the Laboratory Animal Use and Care Committee of Guangzhou Medical University strictly under license number 2018–204 and conformed to the Chinese National Institute of Health Guide for the Care and Use of Laboratory Animals.

### Mice

6–8 weeks old female C57BL/6 mice were purchased from the Animal Experimental Center of Sun Yat-Sen University (Guangzhou, China) and female TLR7-KO mice (B6.129S1-Tlr7tm1Flv/J, strains: 008380) were purchased from the Jackson Laboratory Repository. All mice were maintained in a specific pathogen-free microenvironment at the Laboratory Animal Centre, Guangzhou Medical University.

### Parasites and infection

*S. japonicum* cercariae were shed from naturally infected Oncomelania hupensis snails, which were purchased from the Chinese Institute of Parasitic Disease (Shanghai, China). C57BL/6 mice were infected percutaneously with 40 ± 5 cercariae and sacrificed at some points after infection. Animal experiments were performed in strict accordance with the regulations for the Administration of Affairs Concerning Experimental Animals, and all efforts were made to minimize suffering.

## Reagents and antibodies

RPMI 1640 (Cat:C11875500BT), FBS (Cat:10099141), penicillin-streptomycin (Cat:15140122), 5-(and-6)-chloromethyl-2,7-dichlorodihydro fluorescein diacetate, acetyl ester (CM-H2DCFDA) (Cat:C6827), and CFSE (Cat:C34554) were obtained from Invitrogen (Grand Island, NY). Red blood cell (RBC) lysis buffer (Cat:C3702) was obtained from Beyotime Biotechnology (Shanghai, China). Recombinant murine GM-CSF (Cat:315–03) and IL-6 (Cat:216–16) were purchased from Peprotech (Oak Park, CA). Phorbol 12-myristate 13-acetate (PMA) (Cat:P1585), Brefeldin A (Cat:B5936), Ionomycin (Cat:I3909), ConA (Cat:C2010), R848 (Cat:SML0196) and dimethyl sulfoxide (DMSO) (Cat:D2650) were purchased from Sigma-Aldrich (St. Louis, MO). Fluorescein-conjugated anti-mouse antibodies: CD45-APC (30-F11) (Cat:17-0451-83), CD11b-APC-Cy7 (M1/70) (Cat:47-0112-82), CD11b-PE-Cy7 (M1/70) (Cat:25-0112-82), CD11b-PE-Cy5 (M1/70) (Cat:15-0112-83), Gr1-FITC (RB6-8C5) (Cat:11-5939-86), Gr1-PE (RB6-8C5) (Cat:MA1-83934), Gr1-PE-Cy5 (RB6-8C5) (Cat:15-5939-81), CD3e-PE-Cy7 (145-2C11) (Cat:25-0031-82), CD4-PECP-Cy5.5 (GK1.5) (Cat:15-0041-83), CD8-FITC (53–6.7) (Cat:MAI-10303), CD103-PE (2E7) (Cat:12-1031-83), IFN-γ-APC (XMG1.2) (Cat:17-7311-82), IL-1α-PE (ALF-161) (Cat:12-7011-82), IL-10-PE (JES5-16E3) (Cat:12-7101-82), Ly6C-PECP-Cy5.5 (HK1.4) (Cat:45-5932-82) and their corresponding isotype controls were obtained from eBioscience (San Diego, CA). Fluorescein-conjugated anti-mouse antibodies: CD3-APC-Cy7 (145-2C11) (Cat:100330), TLR3-APC (11F8) (Cat:141906), PD-L1-BV421 (10F.6G2) (Cat:124315), CD115-APC-Cy7 (AFS98) (Cat:135524), CD135-BV421 (A2F10.1) (Cat:135313), GM-CSF-PECP-Cy5.5 (MP1-22E9) (Cat:505409), p-STAT3-BV421 (13A3-1) (Cat:651009) and their corresponding isotype controls were obtained from BioLegend (San Diego, CA). Fluorescein-conjugated anti-mouse antibodies: Ly6G-PE (1A8) (Cat:551461), CD69-BV421 (H12F3) (Cat:562920), TLR7-PE (A94B10) (Cat:565557), PD-L2-APC (TY25) (Cat:560086), IL-6-APC (MP5-20F3) (Cat:561367) and their corresponding isotype controls were obtained from BD Biosciences (San Jose, CA). p-STAT3 (Tyr705) (D3A7) Rabbit mAb (Cat:9145T), p-NF-κB p65 (Ser536) (93H1) Rabbit mAb (Cat:3033T), and Alexa Fluor 488 Anti-rabbit IgG (H+L) F (ab')2 fragment antibodies (Cat:4412S) were purchased from CST (Cell Signaling Technology, Inc.). The eFluor506-FVD (Cat:65-0863-14) was obtained from eBioscience. The AnnexinV-PE Apoptosis Detection Kit (Cat:559763) were obtained from BD Biosciences (San Jose, CA).

## SEA preparation

The SEA of *S. japonicum* used in culture is from Jiangsu Institute of Parasitic Diseases (China). Polymyxin B agarose beads (Sigma) were used to remove endotoxin from aseptic filtered SEA, and Limulus amoebocyte lysate assay kit (Lonza, Switzerland) was used to detect whether endotoxin was removed.

## Histology studies

After obtaining part of the liver and lungs of mice, the organs were perfused with 0.01M -buffered saline (pH = 7.4) for 3 times, fixed in 10% formalin, then embedded in paraffin and sliced. The sections were stained with conventional hematoxylin-eosin (H&E) and observed under light microscope. To quantify the granuloma area, histological images were analyzed using the ImageJ software (National Institutes of Health, Bethesda, MD). Granulomas were measured by an observer blind to treatment history. Granuloma size is expressed as mean area in $\mu m^2 \pm$ SD.

## Pulmonary lymphocyte isolation

After the lungs of mice were removed, the peripheral blood was cleared by lavage with PBS. Then the lung tissue was cut into small pieces with scissors and 30 minutes was digested by adding digestive buffers (including collagenase IV and DNAase I) at 37˚C. The digested lung tissue was ground and crushed with a sterile syringe piston and filtered with a 100 mm filter. Then the cell suspension was treated with RBC lysis buffer for 10 minutes and washed twice with HBSS. Finally, the lymphocytes were resuscitated with complete 1640 culture medium for follow-up experiments.

## Cell surface staining and cell population isolation

Cells were washed twice in PBS and blocked in PBS buffer containing 1% BSA for 30 mins. Then, the cells were stained with conjugated antibodies that were specific for cell surface antigens at 4˚C in the dark for 30 mins. These antigens included FVD, CD45, CD11b, Gr1, CD3, CD4, CD8, CD69, Ly6G, Ly6C, PD-L1, PD-L2, CD135, CD115 and CD103. The stained lymphocytes were analyzed using flow cytometry, and the results were analyzed by CytoExpert 2.3 software (Beckman Coulter Fullerton, CA). Isotype-matched cytokine controls were included in each staining scheme. For the purification of MDSCs, mouse single cell suspension of mouse lung was stained with APC-Cy7-CD11b and PE-Gr1 antibodies and CD11b$^+$Gr1$^+$cells were isolated by cell sorting on a FACS Aria cell sorter (BD, Mountain View, CA). For isolation of T cells, mouse splenocytes were stained with the CD3e-PE-Cy7 antibody and CD3$^+$ T cells were purified by flow cytometric sorting.

## Allogeneic mixed lymphocytes reaction

The proliferation of T cells was detected by CFSE dilution method. Purified CD3$^+$ T cells from BALB/c mice were labeled with CFSE (2μM), stimulated with ConA (5μg/mL) and cultured alone or cocultured with MDSCs for 3 days. Cells were then stained with CD4 or CD8 antibodies, and T-cell proliferation was analyzed by flow cytometry.

## Quantitative real-time PCR (qRT-PCR)

Total RNA of MDSCs were isolated by using Trizol reagent (Invitrogen Life Technologies, Carlsbad, USA), following the manufacturer's protocol, including DNase treatment of the samples. 0.5 μg of total RNA was transcribed to cDNA by using a SuperScript III Reverse Transcriptase Kit (Qiagen, Valencia, CA). The levels of TLR transcripts were normalized to β-actin transcripts by using the relative quantity (RQ) = $2^{-\triangle\triangle Ct}$ method. The sequences of primers were listed in Table 1.

## Cell culture

One million single cell suspension of the mouse lung were cultured in RPMI 1640 medium supplemented with 10% FBS, 20 ng/mL GM-CSF, and 20 ng/mL IL-6. According to the needs

**Table 1. Sequences of primers.**

| Gene | Forward Primer(5'-3') | Reversed Primer(5'-3') |
|---|---|---|
| TLR2 | AAGATGTCGTTCAAGGAGGTGCG | ATCCTCTGAGATTTGACGCTTTG |
| TLR3 | ATTCGCCCTCCTCTTGAACA | TCGAGCTGGGTGAGATTTGT |
| TLR4 | ACCTGGAATGGGAGGACAATC | AGGTCCAAGTTGCCGTTTCT |
| TLR7 | CCACATTCACTCTCTTCATTGG | GGTCAAGAACTTCCAGCCTG |
| β-actin | CCGTAAAGACCTCTATGCCAAC | GGGTGTAAAACGCAGCTCAGTA |

of the experiment, different stimulants R848 (2 ug / ml, 10 ug / ml, 20 ug / ml) and SEA (100 ug / ml) were mixed, and the cultures were maintained at 37˚C in a 5% CO2-humidified atmosphere in 48-well plates. The cells were analyzed by flow cytometry on day 3.

## Cell intracellular cytokine and molecular staining

For intracellular cytokine staining, single cell suspensions from the lungs of control mice and mice infected with *S. japonicum* were stimulated with 20 ng/mL phorbol 12-myristate 13-acetate (PMA) plus 1 μg/mL Ionomycin for 5 h at 37˚C under a 5% CO2 atmosphere. Brefeldin A (10 g/mL, Sigma Shanghai, China) was added during the last 4 h of incubation. Cells were washed twice in PBS, fixed with 4% paraformaldehyde, and permeabilized overnight at 4˚C in PBS buffer containing 0.1% saponin (Sigma), 0.1% BSA and 0.05% NaN3. Cells were then stained for 30 mins at 4˚C in the dark with conjugated antibodies specific for the cell surface antigens CD11b and Gr1 as well as the cytokines GM-CSF, IFN-γ, IL-1α, IL-6 and IL-10. Isotype-matched cytokine controls were included in each staining protocol. Stained cells were washed twice and detected by FCM. For p-NF-κB p65 and p-STAT3 detection, cells were stained with antibodies of cell surface markers first. They were fixed and permeabilized with transcription factor staining buffer set (Invitrogen, 00-5523-00), and cultured with antibodies specific for p-NF-κB p65 and p-STAT3. Then the cells were stained with the second antibodies conjugated with fluorescences.

## ROS production

The ROS produced by the cells can be detected by Oxidation-sensitive dye CM-H2DCFDA (Invitrogen). Cells were incubated at 37˚C in RPMI1640 in the presence of 1 μM CM-H2DCFDA for 30 mins and then labeled with CD11b-PE-Cy7 and Gr1-PE-cy5 antibodies in the dark at 4˚C for 30 mins. The ROS content in MDSCs was detected by flow cytometry.

## Annexin V staining

The single cell suspension of the mouse lung was first stained with MDSCs (CD11b$^+$Gr1$^+$) fluorescence labeled antibody at 4˚C for 30 minutes, then washed with PBS, and then stained with the Annexin-V-PE according to the manufacturer's instructions (BD Pharmingen).

## Statistics

Statistical analyses of the diferences between means were performed using unpaired, two-tailed tests. If the data is non-normally distributed, we used a nonparametric test to compare the diference. The dynamic proportion of CD11b$^+$Gr1$^+$ MDSCs, CD3$^+$CD4$^+$ and CD3$^+$CD8$^+$ T cells in lung tissue were analyzed by single factor analysis of variance. The software packages GraphPad Prism (v6.02) and SPSS Statistics 17.0 were used. $p < 0.05$ was considered to be statistically significant.

## Results

### *S. japonicum* infection induces MDSCs in the mouse lung

Mice were sacrificed 6–7 weeks after infection with *S. japonicum*, the livers and lungs were harvested, and sections were made as described in the Materials and Methods. The H&E staining results showed that lesions and granuloma were observed in the infected mouse livers (Fig 1A and 1B), pulmonary interstitial thickening and leukocyte infiltration (Fig 1C and 1D). To explore the existence and distribution of MDSCs in the lung of *S. japonicum* infected mice at 6–7 weeks after infection, mono-nuclear cells were isolated from the mouse lung, and the

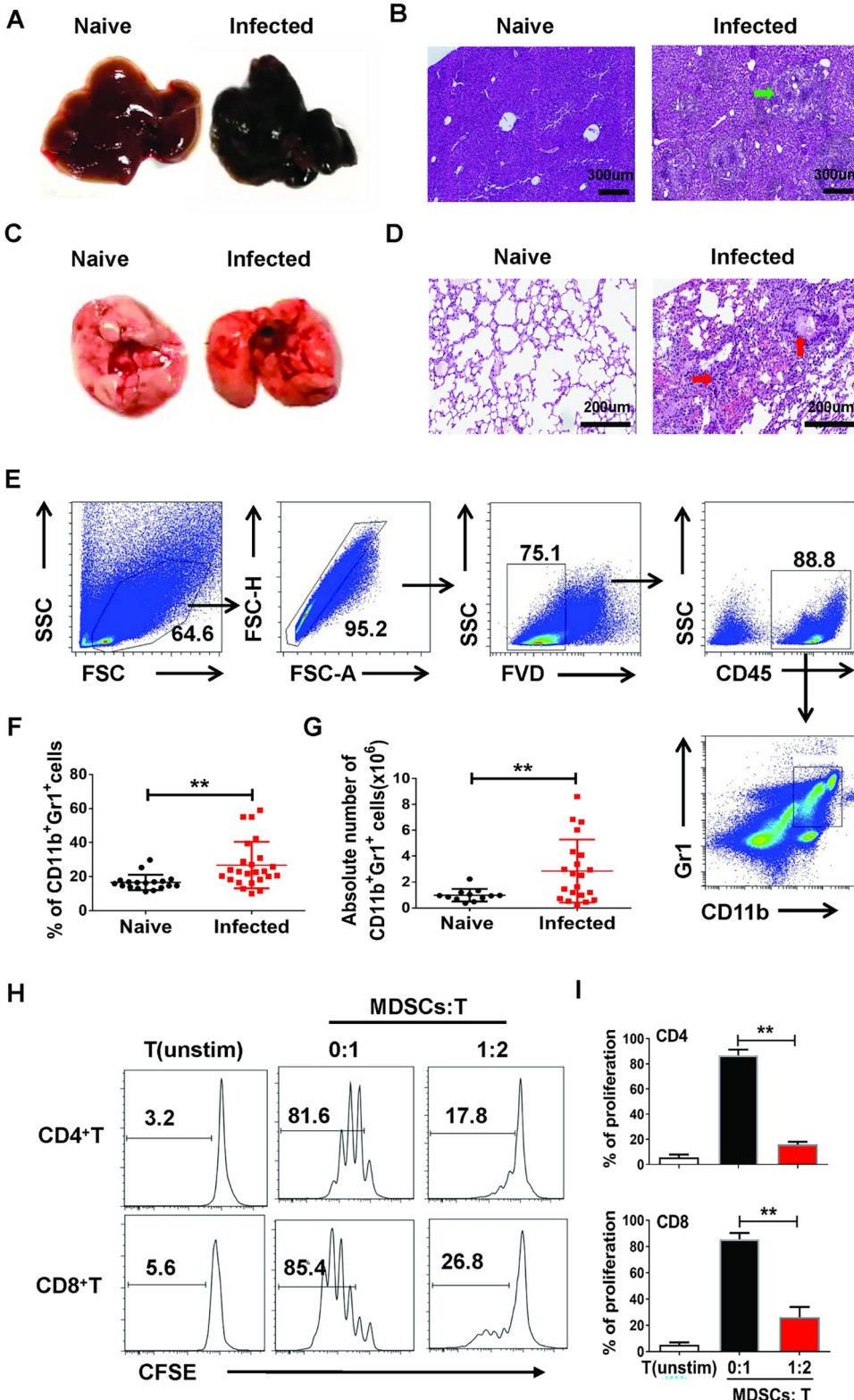

**Fig 1. *S. japonicum*-infection promoted MDSCs accumulation in the mouse lung.** (A-D) C57BL/6 mice were infected percutaneously with 40 ± 5 cercariae and sacrificed at 6–7 weeks after infection; Liver and lung tissues were harvested. (A) Representative images of livers. (B) Representative images of liver H&E staining; the arrow indicates granuloma. (C) Representative images of lungs. (D) Representative images of lung HE staining: the right arrow

indicates leukocyte infiltration and the upward arrow indicates interstitial thickening of the alveolar cavity. (E) Gated strategy of MDSCs: all diploid cells, dead cells and non-lymphoid cells were excluded in this study. (F-G) The percentage (F) and absolute numbers (G) of CD11b$^+$Gr1$^+$ cells in lungs of normal and infected mice were evaluated by flow cytometry after staining with specific antibodies. Data are expressed as the mean ± SD of 12–24 mice from four independent experiments, $^*$ $p < 0.05$, $^{**}$ $p < 0.01$, compared with the corresponding control, unpaired $t$-test was used. (H-I) MDSCs was sorted and purified by flow cytometry, and then co-cultured with ConA-stimulated CD3$^+$ T cells at 1:2 ratios for 3 days. T cell proliferation was measured by CFSE dilution. Unstimulated T cells were used as negative control. Three independent experiments were performed and showed similar results, and the mean ± SD of six samples pooled from the three experiments is shown. $^*$ $p < 0.05$, $^{**}$ $p < 0.01$, compared with the controls, unpaired $t$-tests were used.

percentage of CD11b and Gr1 co-expressed MDSCs were detected by flow cytometry (Fig 1E). The results demonstrated that the percentage and absolute numbers of CD11b$^+$Gr1$^+$ MDSCs in the lungs of infected mice were higher than those in normal mice (p < 0.05, Fig 1F and 1G). Furthermore, to further explore the function of *S. japonicum* infection-induced pulmonary MDSCs, lymphocytes were isolated from *S. japonicum* infected mouse lung. FACS was used to sort MDSCs, which were then co-cultured with ConA pre-stimulated, CFSE-stained T cells from BALB/c mice at a 1:2 ratio. Three days later, the proliferation of T cells was analyzed by FACS. The results indicated that *S. japonicum* infection-induced pulmonary MDSCs could inhibit the proliferation of both CD4$^+$ and CD8$^+$ T cells significantly at the ratio of MDSCs:T was 1:2 (p < 0.05, Fig 1H and 1I).

## Changes of MDSCs and T cells in the lung of mice during infection of *S. japonicum*

To further explore the effect of *S. japonicum* infection-induced MDSCs on T cells in the lungs of the infected mice. We investigated the dynamic changes of pulmonary MDSCs, CD3$^+$CD4$^+$ T cells and CD3$^+$CD8$^+$ T cells from week 2 to week 7 of *S. japonicum*-infected mice. As shown in Fig 2A–2D, the percentage and the absolute number of pulmonary MDSCs were slightly decrease in the week 4 after *S. japonicum* infection, but remarkably increased in the course of infection and maintained at a high level from week 6 to week 7 (Fig 2A and 2B). The percentage of CD3$^+$CD4$^+$ T cells and CD3$^+$CD8$^+$ T cells increased at first and then decreased, reaching the highest value at the week 3 and week 4, and then decreasing gradually, and the lowest at the 7 week, which was opposite to the trend of MDSCs (Fig 2C and 2D). In addition, the expression of CD69 molecules associated with T cell activation in CD3$^+$CD4$^+$ T cells and CD3$^+$CD8$^+$ T cells was also opposite to the trend of MDSCs during infection of *S. japonicum* (*p* < 0.05, Fig 2E and 2F). These results suggested that *S. japonicum* infection-induced pulmonary MDSCs could inhibit immune response by inhibiting the proliferation and activation of CD3$^+$CD4$^+$ T cells and CD3$^+$CD8$^+$ T cells in the lungs of mice during infection of *S. japonicum*.

## TLR7 modulating MDSCs expansion in the lung of mice in vitro and in vivo

To further confirm the cause of the increased MDSCs in lung of *S. japonicum* infected mouse, we used qRT-PCR to detect the expression of Toll-like receptors (TLR2, TLR3, TLR4, TLR7) in pulmonary MDSCs from normal mice (Naive MDSCs) and infected mice (Inf MDSCs). Compared with Naive MDSCs, the expression of TLR2, TLR3 and TLR7 in Inf MDSCs had an upward trend and the expression of TLR7 increased most significantly, while the expression of TLR4 decreased (Fig 3A). In addition, TLR7 had the highest level of expression in Inf MDSCs from *S. japonicum* infected mouse lung compared with TLR2, TLR3 and TLR4 (Fig 3A).

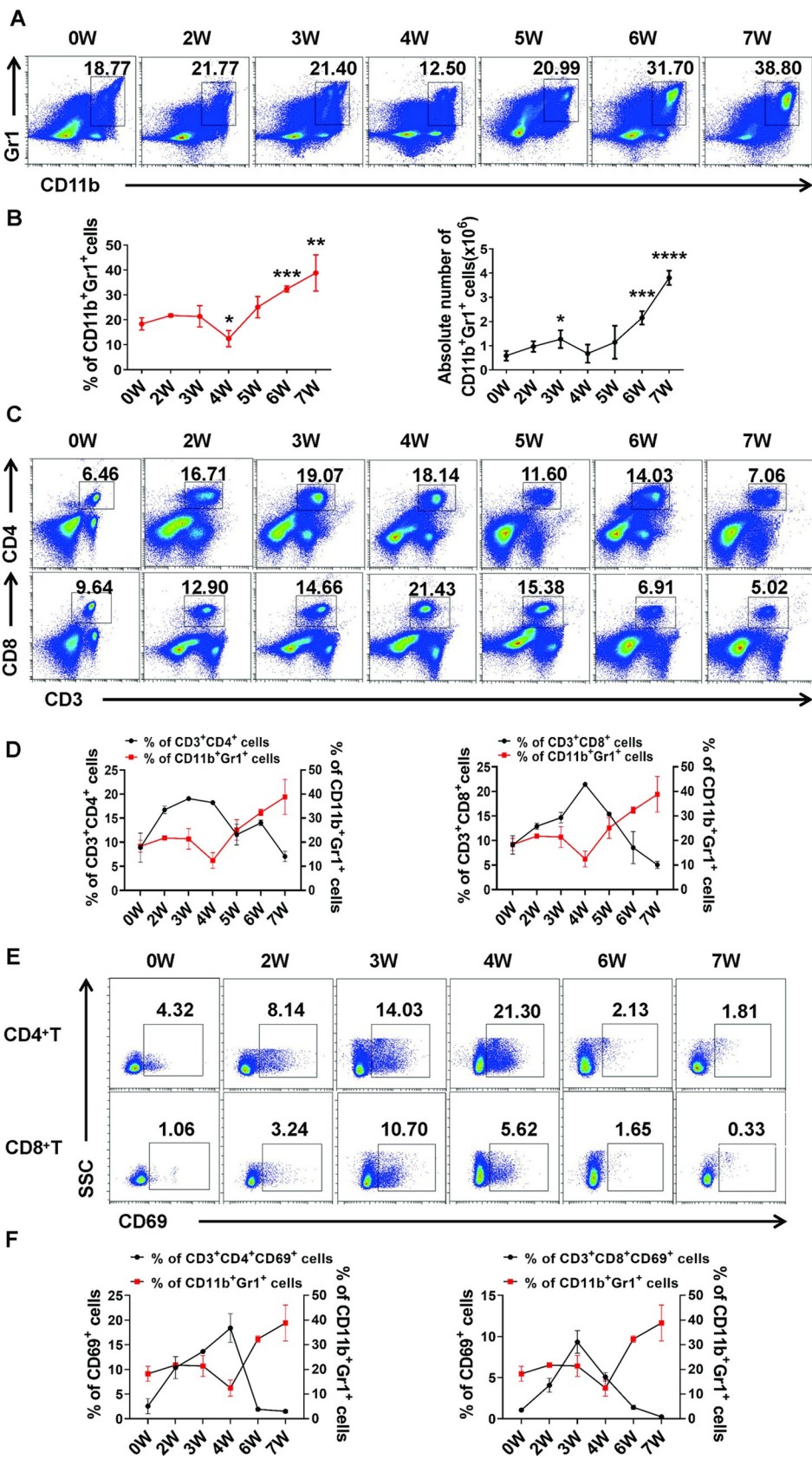

**Fig 2. Dynamic changes of MDSCs and T cells in the lung of *S. japonicum* infected mice.** (A-F) Dynamic changes of CD11b$^+$Gr1$^+$ MDSCs (A-B), CD3$^+$CD4$^+$ T cells, CD3$^+$CD8$^+$ T cells (C, D) and CD69 expression in CD3$^+$CD4$^+$ T cells and CD3$^+$CD8$^+$ T cells (E, F) in infected mice in mouse lung from week 2 after infection were determined by FCM; naive control was set (0 week), compared with naive groups. (A, C, E) Representative results of three independent results are shown. (B) The percentage and absolute number of MDSCs (CD11b$^+$Gr1$^+$ cells) were analyzed by cell surface staining. (D, F) Quantification of CD11b$^+$Gr1$^+$ MDSCs in lung across the days of infection (as described in B, red line), compared to the percentage of CD3$^+$CD4$^+$ T cells, CD3$^+$CD8$^+$ T cells (D) and CD69 expression in CD3$^+$CD4$^+$ T cells and CD3$^+$CD8$^+$ T cells (F) (black line). Each data point represents the mean ± SD of 3–10 mice. (B, D, F) Different groups of mice were killed and analyzed on each day of infection. Significance determined by unpaired Student's *t*-test $^*$ $p < 0.05$, $^{**}$ $p < 0.01$.

Furthermore, we used flow cytometry to detect the expression of TLR3 and TLR7 in three groups of cells in the mouse lung: CD11b$^+$Gr1$^+$, CD11b$^+$Gr1$^-$, and CD11b$^-$Gr1$^-$ cells. As shown in the Fig 3B, the expression of TLR3 and TLR7 in CD11b$^+$Gr1$^+$ cells was higher than that of the other two groups of cells ($p < 0.001$), and the expression of TLR7 in the three groups of cells was significantly more than that of TLR3. These results suggested that MDSCs in the lungs of mice may be regulated by TLR7.

To confirm that TLR7 has a regulatory effect on pulmonary MDSCs, single cell suspension from the lungs of normal C57BL/6 mice was cultured with different concentrations of TLR7 agonist R848, and the percentage of MDSCs was detected by flow cytometry 3 days later. The results showed that the percentage of MDSCs in the group treated with different concentrations of R848 was higher than that in the control group ($p < 0.05$, Fig 3C). Then, to further explore the roles of SEA and R848 in *S. japonicum* infection-induced pulmonary MDSCs, lung single cell suspension from normal C57BL/6 mice was isolated and stimulated with GM-CSF and IL-6 for 3 days. SEA and R848 were added to the cells alone or together, and a negative control was used as described in Materials and Methods. 3 days later, the percentages of MDSCs were detected by FACS. The results indicated that both SEA and R848 could significantly increase the percentages of MDSCs ($p < 0.05$), but the effects of SEA and R848 could not be overlaid and R848 could down-regulate the induction of SEA on MDSCs ($p < 0.05$, Fig 3D). The percentages of MDSCs in the lungs of WT and TLR7-KO mice infected with or without *S. japonicum* were further investigated by flow cytometry. Interestingly, the results indicated that TLR7-KO could decrease the percentage of MDSCs in the absence of infection ($p < 0.05$), but could dramatically increase the percentage of MDSCs infected with *S. japonicum*. ($p < 0.05$, Fig 3E). These results suggested that TLR7 might have different effects on MDSCs under normal and infection conditions.

## Effect of TLR7 on the phenotype and function of MDSCs in the lung during *S. japonicum* infection

To further explore the role of TLR7 on pulmonary MDSCs in mice infected with *S. japonicum*, the subsets, activation and functional molecules of pulmonary MDSCs were detected in both normal and *S. japonicum* infected WT and TLR7-KO mice. Although the percentages of CD11b$^+$Ly6G$^+$Ly6C$^{-/low}$ PMN-MDSCs were significantly increased ($p < 0.05$) and CD11b$^+$Ly6G$^-$ Ly6C$^{high}$ M-MDSCs were significantly decreased in lung of infected mice compared to naive mice ($p < 0.05$), there was no difference in the percentages of CD11b$^+$Ly6G$^+$Ly6C$^{-/low}$ PMN-MDSCs and CD11b$^+$Ly6G$^-$Ly6C$^{high}$ M-MDSCs ($p > 0.05$) between infected WT mice and infected TLR7-KO mice (Fig 4A). The percentages of PD-L1 and PD-L2 expressing MDSCs from infected TLR7-KO mice were higher than those from infected WT mice ($p < 0.05$), but the expression of CD115, CD135 and CD103 was not significantly different between infected WT mice and infected TLR7-KO mice ($p < 0.05$, Fig 4B).

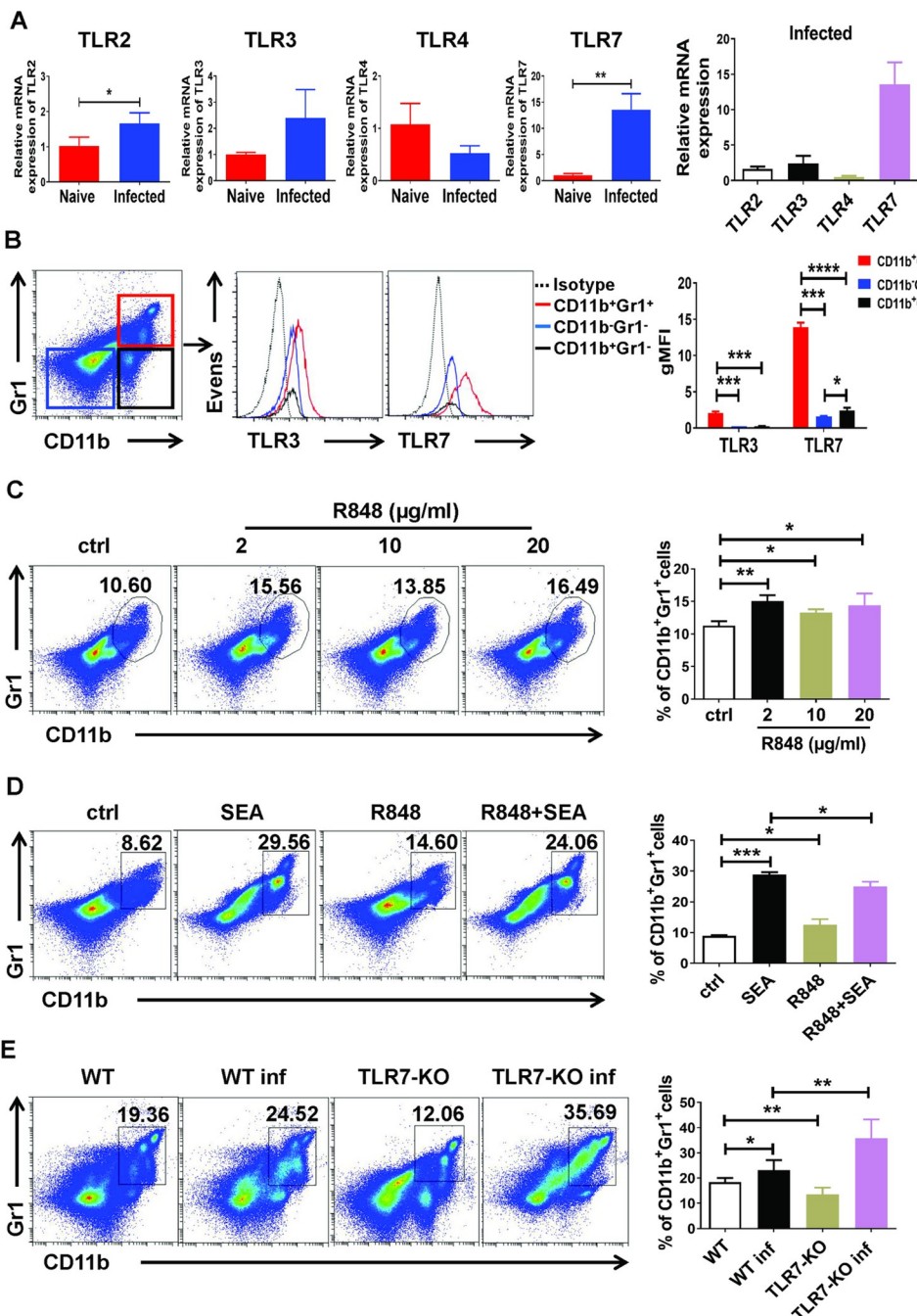

**Fig 3. TLR7 modulating MDSCs expansion in the lung of mice.** (A) In pulmonary CD11b⁺Gr1⁺ cells purified from normal mice (Naive MDSCs), and pulmonary CD11b⁺Gr1⁺ MDSCs purified from infected mice (Inf MDSCs), gene expression was determined by qRT-PCR. (B) The single cell suspension of normal mouse lung was isolated and the fluorescence expression intensity of TLR3 and TLR7 in group CD11b⁺Gr1⁺, CD11b⁺Gr1⁻, CD11b⁻Gr1⁻ cells was detected by flow cytometry. Data are expressed as the mean ± SD of 3 mice. (C-D) Mouse single cell suspension of lung was cultured in GM-CSF (20 ng/mL) for 3 days in the presence of R848 at different concentrations (C), 100 ug/mL SEA or/and 2 ug/mL R848 (D); the vehicle was used as the control. The proportions of MDSCs (CD11b⁺Gr1⁺ cells) were evaluated by flow cytometry. The results from a single experiment (left), as well as the mean ± SD of three independent experiments (right), are shown. (E) C57BL/6 mice and TLR7⁻/⁻ mice were infected percutaneously with 40 ± 5 cercariae and sacrificed at 6–7 weeks after infection, the lung single cell suspensions of WT (C57BL/6), WT inf (C57BL/6 infected), TLR7-KO (TLR7⁻/⁻) and TLR7-KO inf (TLR7⁻/⁻ infected) mice were isolated. The percentage of MDSCs (CD11b⁺Gr1⁺ cells) was detected by flow cytometry. Data are expressed as the mean ± SD of 6–10 mice from four independent experiments (A-E). * $p < 0.05$, ** $p < 0.01$ compared with the corresponding control, unpaired *t*-test was used.

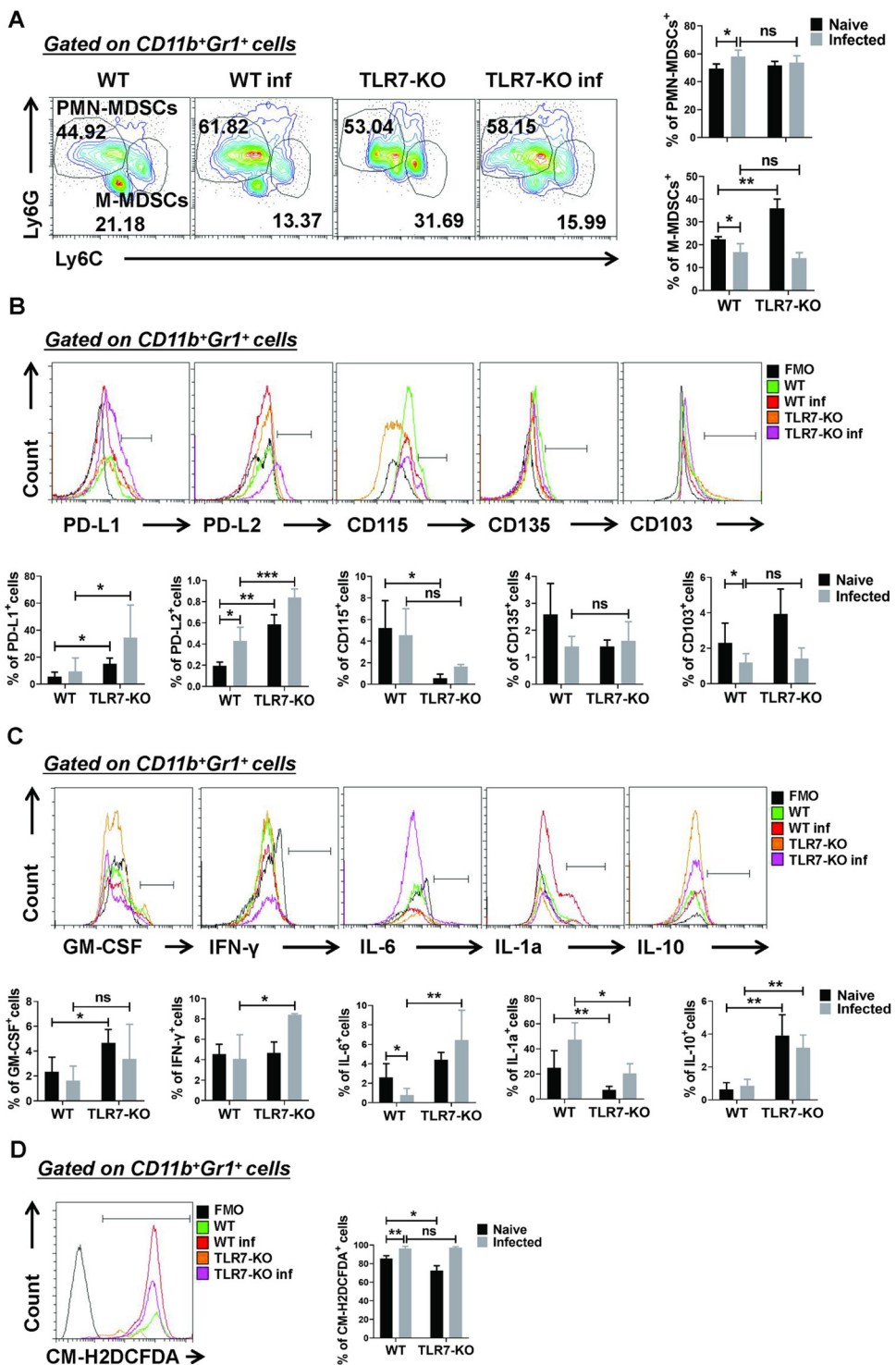

**Fig 4. Effect of TLR7 on the phenotype and function of MDSCs in the lung during *S. japonicum* infection.** (A-D) C57BL/6, TLR7[-/-] mice were infected with 40 ± snail eggs through the skin, and lung single-cell suspensions were isolated from naive and infected WT and TLR7-KO mice after 6–7 weeks for flow cytometry. (A) The proportions of the MDSCs subtypes in the lungs were evaluated by flow cytometry. Representative results (left) and the statistical graph (right) are shown. Data are expressed as the mean ± SD of 3–6 mice. (B) Phenotypic analysis of CD11b[+]Gr1[+] cells from the lungs of mice by different fluorescence-labeled Abs to mouse surface markers, including PD-L1, PD-L2, CD115, CD135, and CD103. Representative results (up) and statistical graph (down) are shown. Data are expressed as the mean ± SD of 3–9 mice. (C) Single cell suspensions of lung cells from mice were stimulated with PMA and

 

ionomycin. The expression of GM-CSF, IFN-γ, IL-1α, IL-6 and IL-10 were detected in MDSCs by FACS analysis. Data are expressed as the mean ± SD of 3–7 mice. A representative of two independent experiments is shown. (D) MDSCs were harvested from the lung and the ROS level was evaluated by flow cytometry. CD11b⁺Gr1⁺ cells were gated and the percentage of CM-H2DCFDA⁺ cells is shown as the mean ± SD of 6 samples pooled from three independent experiments. Data are expressed as the mean ± SD of 3 mice. (A-D) $^*$ $p < 0.05$, $^{**}$ $p < 0.01$ compared with the corresponding control, unpaired $t$-test was used.

Next, intracellular cytokine staining and the expression levels of ROS were performed on pulmonary MDSCs isolated from different groups of mice. The results indicated that IFN-γ, IL-6 and IL-10 secreted by pulmonary MDSCs from infected TLR7-KO mice significantly increased compared to infected WT mice ($p < 0.05$), while IL-1α⁺ MDSCs from the lungs of infected TLR7-KO mice significantly decreased compared to infected WT mice ($p < 0.05$, Fig 4C). Although the expression levels of ROS significantly increased in the lungs of infected mice compared to naive mice ($p < 0.05$), there was no difference in the expression levels of ROS ($p > 0.05$, Fig 4D) between infected WT mice and infected TLR7-KO mice. Taken together, these results indicated that TLR7 could regulate pulmonary MDSCs activation and function in the course of *S. japonicum* infection.

## Effect of TLR7 on MDSCs are dependent on the activation of NF-κB p65

To further explore the mechanism of the effects of TLR7 on pulmonary MDSCs, the apoptosis and activation of STAT3 and NF-κB p65 in MDSCs from the lungs of infected or uninfected WT and TLR7-KO mice were investigated by flow cytometry. As shown in Fig 5A and 5B, there was no difference in the percentages of Annexin V⁺ MDSCs between four groups. The proportions of STAT3 and NF-κB p65 phosphorylated MDSCs were increased in the infected mice compared with the uninfected mice ($p < 0.05$). The proportions of STAT3 phosphorylated MDSCs from TLR7-KO infected mice were decreased compared with those from the WT infected mice ($p < 0.05$), while the proportions of p65 phosphorylated MDSCs from TLR7-KO infected mice were increased compared with that from the WT infected mice ($p < 0.05$). Similarly, as shown in Fig 5C and 5D, the proportions of STAT3 and NF-κB p65 phosphorylated MDSCs in SEA stimulated WT and TLR7-KO cells increased ($p < 0.05$). The percentage of Annexin V⁺ MDSCs was sincerely increased after TLR7 deficiency. The proportions of STAT3 and p65 phosphorylated MDSCs from SEA stimulated TLR7-KO cells were increased compared with those from the WT cells ($p < 0.05$). These results indicated that TLR7 could regulate pulmonary MDSCs through negatively regulating TLR7–MyD88–NF-κB pathway in the course of *S. japonicum* infection.

## Effect of TLR7 on pathological changes of the lung after *S. japonicum* infected

To elucidate the role of TLR7 on lung after *S. japonicum* infection, both WT and TLR7-KO mice were infected at the same time. Mice were euthanized 6–7 weeks after infection, and lungs were removed. As shown in Fig 6A, compared with infected WT mice, the size of the lungs in infected TLR7-KO mice was similar. However, the color was paler. As shown in Fig 6B and 6C, the weight of the lung from infected TLR7-KO mice was higher than that from the infected WT mice ($p < 0.05$), but there was no obvious difference in the number of lung cells between the infected WT and infected TLR7-KO mice. As shown in Fig 6D, lung biopsy (HE staining) was observed, and the structure of lung tissue in naive mice was clear with a uniform distribution of lung cells, but alveolar fusion and inflammatory cell aggregation were found in the lung tissue of infected and knockout mice. Compared with WT infected mice, TLR7-KO

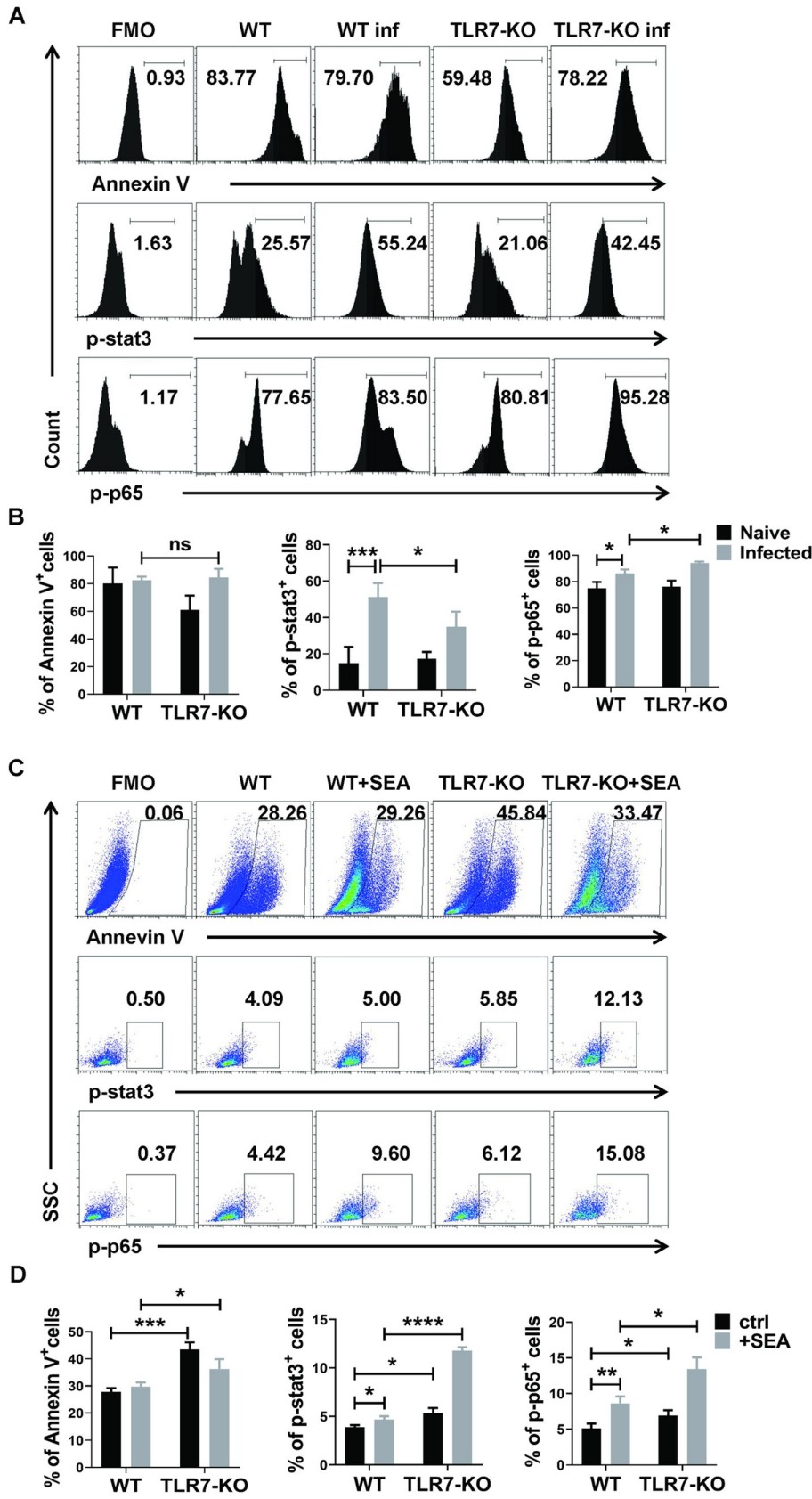

**Fig 5. The effects of TLR7 on MDSCs are dependent on the activation of NF-κB p65.** (A, B) C57BL/6, TLR7$^{-/-}$ mice infected with 40 ± 5 snail eggs through skin. Lung single cell suspensions were isolated from naive and infected WT and TLR7-KO mice after 6–7 weeks for flow cytometry detection. Data are expressed as the mean ± SD of 3–5 mice. (C, D) C57BL/6 and TLR7$^{-/-}$ mouse lungs single cell suspension were cultured in GM-CSF (20 ng/mL) for 3 days in the presence of 100 ug/mL SEA, the vehicle was used as the control. The proportions of Annexin V$^+$, p-stat3$^+$, and p-p65$^+$ MDSCs (CD11b$^+$Gr1$^+$ cells) were evaluated by flow cytometry. The results from a single experiment (up), as well as the mean ± SD of three independent experiments (down), are shown. $^*$ $p < 0.05$, $^{**}$ $p < 0.01$, compared with the corresponding control, unpaired *t*-test was used.

infected mice had more alveolar fusion, thickened alveolar cavity and more severe damage. Moreover, the area of granuloma was calculated as described in the section Materials and Methods. As shown in Fig 6E, the area of granuloma in lung tissue was significantly increased in infected TLR7-KO mice ($p < 0.05$). These results indicated that the absence of TLR7 aggravates the lung damage in the mice infected with *S. japonicum*.

## Discussion

Schistosomiasis is a zoonotic parasitic disease that seriously affects human health. Many adults and children with schistosomiasis develop lung symptoms such as coughing, chest pain and bloody sputum. In addition to their respiratory and metabolic functions, the lungs also play a role in the immune system. The role of MDSCs in lung during a *S. japonicum* infection has not been studied, even though MDSCs contribute to many infectious diseases. This study provides evidence that TLR7 plays an important role in regulating MDSCs in lung diseases caused by *S. japonicum* infection.

MDSCs are important new myeloid suppressor cells that can be induced in different types of pathogenic microorganisms, including parasitic infections [34]. In this study, the percentage and the absolute numbers of CD11b$^+$Gr1$^+$ MDSCs in the lung of *S. japonicum*-infected mouse were significantly increased (Fig 1E–1G), which suggested that *S. japonicum* infection could induce MDSCs accumulation in the lungs of mice. Consistent with our results, MDSCs were reported to accumulate in the BM, spleen and lymph nodes of *S. japonicum*-infected mice [24]. It is well-known that MDSCs are a population of heterogeneous immature myeloid cells, and one of their principal functions are to inhibit T cell functions [15]. Clearly, our study showed that *S. japonicum* infection-induced pulmonary MDSCs inhibited the proliferation of both CD4$^+$ T cells and CD8$^+$ T cells (Fig 1H). Furthermore, the proportion and activation of CD4$^+$ T cells and CD8$^+$ T cells decreased significantly with the increase of MDSCs at 6–7 weeks after infection (Fig 2). These results suggested that pulmonary CD11b$^+$Gr1$^+$ cells from *S. japonicum*-infected mice have a suppressive function and are considered to be MDSCs [16,35].

Many kinds of immune cells recognize pathogen associated molecular patterns (PAMPs) of schistosoma and modulate the immune response by expressing TLRs [26,33]. Some of TLRs were found expressed on MDSCs during the immune response induced by invaded pathogens [32,36]. To explore the effects of TLRs on pulmonary MDSCs in the course of *S. japonicum* infection, we compared the expression of TLR2, TLR3, TLR4 and TLR7 in CD11b$^+$Gr1$^+$ cells sorting from naive and infected mice lung. Among these TLRs, the expression of TLR7 increased significantly in pulmonary CD11b$^+$Gr1$^+$ MDSCs, while the expression of TLR2, TLR3 and TLR4 showed no significant difference after infection (Fig 3A). Furthermore, compared with the expression of TLR7 in CD11b$^+$ Gr1$^-$ cells and CD11b$^-$Gr1$^-$ cells, the expression of TLR7 was highest in CD11b$^+$ Gr1$^+$ cells from naive mice lung (Fig 3B). It suggested that TLR7 might be the main factor regulating the accumulation of MDSCs in the course of *S. japonicum* infection, that is why we investigated TLR7 on pulmonary MDSCs. Moreover, TLR7/8

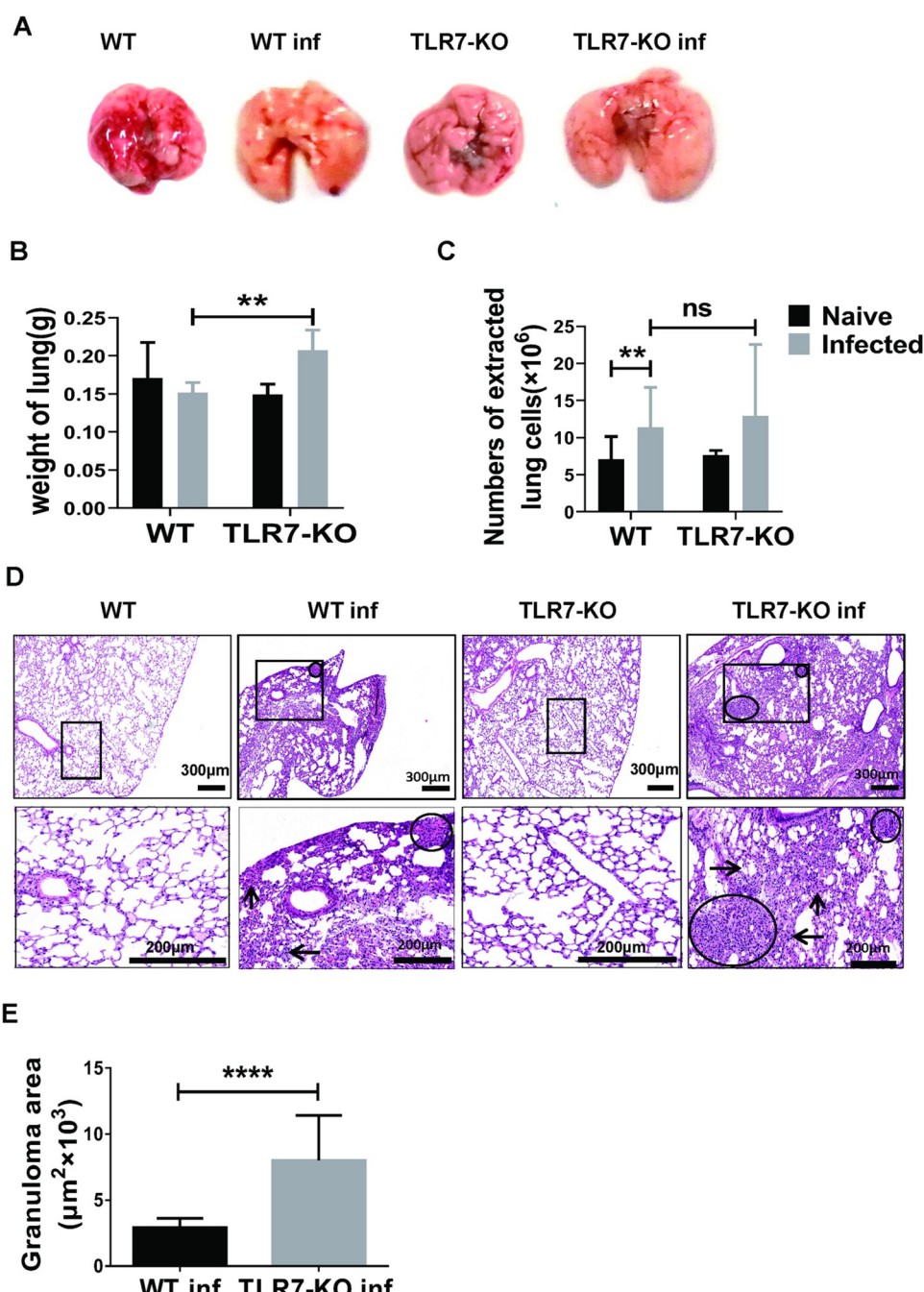

**Fig 6. Pathological changes in the lung of infected TLR7-KO mice.** (A-E) C57BL/6 and TLR7[-/-] mice were infected with 40 ± 5 snail eggs *S. japonicum* cercariae per mouse. 6–7 weeks after infection, mice were euthanized. (A) The representative pictures of the lungs of naive and infected WT and TLR7-KO mice were compared. (B-C) The lung weight and the number of single cell suspension cells in lungs of the four groups of mice were compared. Data are expressed as the mean ± SD of 3–25 mice. (D) The representative picture of HE staining in lungs of the four groups of mice: the left arrow indicates inflammatory cells infiltration, the upward arrow indicates interstitial thickening, the right arrow indicates alveolar dilation, and circle indicates granulomas. Scale bar, 300μm (upper), 200μm (lower). (E) The area of granuloma in infected WT and TLR7-KO mice was compared. A representative of three independent experiments with 5–6 mice per group is shown. Data are expressed as the mean ± SD of 13 to 19 granulomatous areas. * $p < 0.05$ and $p < 0.01$ compared with the corresponding control group, unpaired *t*-test was used.

agonist R848 can induce the development of MDSCs in vitro (Fig 3C). However, R848 did not up-regulate the induction of SEA on MDSCs, but down-regulated it (Fig 3D). Correspondingly, compared with WT mice, the proportion of MDSCs was decreased in uninfected TLR7-KO mice, while the proportion of MDSCs increased significantly in infected TLR7-KO mice (Fig 3E). These results indicated that TLR7 might have different effects on MDSCs under normal and infection conditions, and the mechanism is still unclear [31,37]. The possible reason is that the activation of TLR7 signaling pathway may interact with other signaling pathways in different ways under normal and infection conditions, resulting in different effects of TLR7 on the differentiation and function of MDSCs. Further experiment is needed to elucidate it.

It has been reported that *S. japonicum* infection could specifically induce the increase of PMN-MDSCs in mice spleen [24]. Consistent with our results, we found that the percentage of $CD11b^+Ly6G^+Ly6C^{-/low}$ PMN-MDSCs increased significantly, but the change of $CD11b^+Ly6G^-Ly6C^{high}$ M-MDSCs was decreased in mice lung after infection. However, we found that the differences in the percentage of $CD11b^+Ly6G^+Ly6C^{-/low}$ PMN-MDSCs or $CD11b^+Ly6G^-Ly6C^{high}$ M-MDSCs between WT and TLR7-KO mouse lung were not obvious after infection (Fig 4A), which indicated that TLR7 knockout did not affect the subsets of *S. japonicum* infection-induced lung MDSCs.

In the tumor model, PD-L1/2 was induced in myeloid cells, including MDSCs, which could bind to PD-1 on the activated T cells, and induce cytotoxic T cells exhaustion [19,20]. In this study, we showed that TLR7 deficiency can significantly increase the percentage of PD-L1 or PD-L2-expressed MDSCs in mouse lung after infection (Fig 4B), which indicated that TLR7 deficiency promoted the immunosuppressive function of *S. japonicum* infection-induced MDSCs by up-regulating the expression of PD-L1 or PD-L2 in mouse lung after infection. In addition, MDSCs could secrete a variety of cytokines in pathological conditions to play an immunoregulatory role [18,19]. We compared the percentage of $GM-CSF^+$, $INF-\gamma^+$, $IL-6^+$, $IL-1a^+$, and $IL-10^+$ cells in pulmonary MDSCs from naive and infected WT and TLR7-KO mice to assess inflammatory cytokines production in *S. japonicum* infection induced MDSCs in the mouse lung. Among these cytokines, IL-6 and IL-10 secreted by MDSCs from infected TLR7-KO mice significantly increased compared to infected WT mice (Fig 4C), which suggested that TLR7 knockout might promote the inhibitory function of *S. japonicum* infection-induced pulmonary MDSCs via the secretion of IL-10 in mouse lung and play a pro-inflammatory role by secreting IL-6. In addition, it has been reported that *S. japonicum* infection could induce the expression of the NOX2 subunits $gp91^{phox}$ and $p47^{phox}$, which caused an increase of ROS in MDSCs and mediated the suppressive effects of splenic MDSCs [24]. However, in this study, ROS in pulmonary MDSCs from infected TLR7-KO mice had no obvious difference compared to infected WT mice (Fig 4D). These results confirmed that TLR7 deficiency could promote the function of pulmonary MDSCs by up-regulating the expression of PD-L1 or PD-L2 and secreting of IL-10 in the course of *S. japonicum* infection.

Previous studies have shown that MDSCs apoptosis or myeloid cell differentiation can affect the accumulation of MDSCs [16,38]. To evaluate the effects of these factors on pulmonary MDSCs, we detected the apoptosis of pulmonary MDSCs induced by *S. japonicum* infection. As shown in Fig 5A, there was no significant difference on the expression of Annexin V in pulmonary MDSCs from infected TLR7-KO mice compared to that from infected WT mice, which suggested that TLR7 did not affect the apoptosis of pulmonary MDSCs, and was not responsible for the increase of MDSCs in *S. japonicum* infected mouse lung. NF-κB is the most classical transcription factor regulating myeloid cell differentiation and has been reported to regulate the differentiation and function of MDSCs [39,40]. It is also reported that the JAK2/STAT3 pathway could be activated in MDSCs development [16,24]. The activation of p-

STAT3 and p-p65 were detected in this study (Fig 5A). And the activation of NF-κB p65 was found to be essential for the effects of TLR7 on MDSCs development in the course of *S. japonicum* infection. These results help to clarify the NF-κB pathway is involved in *S. japonicum* infection-induced expansion of pulmonary MDSCs.

TLR7 is an endosomal TLR that recognizes single-stranded RNA (ssRNA). The reason of SEA instead of ssRNA to culture system in our study maybe as follows: first of all, in our study, MDSCs could respond to SEA via TLR7 (Figs 3D and 5C). Secondly, SEA released by mature eggs in the body is a mixture. The mixture is likely to contain substances such as exosomes, which contain complex RNA and proteins. In fact, our SEA was extracted from the in vitro lysis of *Schistosome* eggs, which is a crude extract mixture. It is inevitable to have single-stranded and double-stranded RNA inside. Nonetheless, the effect of ssRNA from *S. japonicum* on pulmonary MDSCs needs to be further investigated.

Although TLR7 deficiency can significantly increase the percentage of PD-L1 or PD-L2-expressed MDSCs in mouse lung after infection (Fig 4B), which indicated that TLR7 deficiency promoted the immunosuppressive function of *S. japonicum* infection-induced MDSCs by up-regulating the expression of PD-L1 or PD-L2 in mouse lung after infection. However, MDSCs have been described as CD11b+Gr1+ immature myeloid cell populations derived from mono-cytes and polymorphonuclear granulocytes that migrate from the blood to the site of infection [15,21]. MDSCs are recruited to the site of inflammation in several models of inflammation [22,41,42]. In this study, TLR7-KO could dramatically increase the percentage of MDSCs infected with *S. japonicum* (Fig 3E) and the lungs of infected TLR7-KO mice showed more infiltration of inflammatory cells, interstitial dilation and granuloma compared with infected WT mice (Fig 6D and 6E), indicating that TLR7 could inhibit the pulmonary inflammation caused by *S. japonicum* infection. Consistent with us, TLR7-KO mice accumulate increased numbers of pulmonary CD11b+Gr1+ MDSCs and show a more intense inflammation in the lung tissues during IAV infection compared to wild type mice [32]. On the other hand, TLR7 knockdown caused serious splenomegaly in infected mice, according to our previous studies [33]. It suggested TLR7 could inhibit the inflammation response in the spleens of *S. japonicum* infected mice. Consistent with us, TLR7 signaling was reported to orchestrate inflammation and innate immunity in response to EV71 infection [43].

Furthermore, it has been previously shown that the presence of MDSCs skews the immune response towards a Th2 response [32,44,45]. This is mainly attributed to their ability to induce IL-10 and reactive oxygen species [32,45,46]. TLR7-KO MDSCs not only produced IL-10, but also IL-6 in mice lung after *S. japonicum* infection (Fig 4C). Interestingly, IL-6 has been shown to amplify the Th2 response [47]. The ability of TLR7-KO MDSCs to coproduce IL-10 and IL-6 may be one mechanism that encourages the Th2 bias observed in TLR7-KO mice during *S. japonicum* infection. It suggested that TLR7 deficiency might increase MDSCs accumulation and Th2 biased response to *S. japonicum* infection in the lungs of mice, which might be related to pulmonary lesions in infected TLR7-KO mice [32,48].

Moreover, the mechanism of immune response induced by *S. japonicum* is very complex: non-specific immunity and specific immunity are mutually conditional and complementary; humoral and cellular immunity regulate and balance each other; interaction between schisto-soma antigens and host MHC; a variety of immune cells, including macrophages, NK cells, B cells and T cells, are involved in the pulmonary lesions in schistosomiasis, and these factors may affect the pathogenesis of schistosomiasis [49–51]. Therefore, in the lungs of infected mice, TLR7-KO may affect the immune response of other immune cells, such as B cells and NK cells, and jointly affect the pulmonary lesions of infected mice [26,33]. This needs to be further studied.

Together, in this study, our findings suggest that TLR7 signaling inhibits the accumulation and function of MDSCs in *S. japonicum* infected mouse lung by down-regulating the expression of PD-L1/2 and secreting of IL-10, via NF-κB signaling.

## Supporting information

**S1 Data. Excel spreadsheet containing, in separate sheets, the underlying numerical data and statistical analysis for Figure panels 1f, 1g, 1i, 2b, 2d, 2f, 3a, 3b, 3c, 3d, 3e, 4a, 4b, 4c, 4d, 5b, 5d, 6b, 6c, and 6e.**
(XLSX)

## Author Contributions

**Conceptualization:** Quan Yang.

**Data curation:** Lu Zhou, Yiqiang Zhu, lengshan Mo, Mei Wang, Jie Lin.

**Formal analysis:** Lu Zhou, Yiqiang Zhu, lengshan Mo, Yi Zhao, Quan Yang.

**Funding acquisition:** Quan Yang.

**Investigation:** Lu Zhou, Mei Wang, Yi Zhao.

**Methodology:** Yuanfa Feng, Anqi Xie, Haixia Wei, Huaina Qiu.

**Project administration:** Jun Huang, Quan Yang.

**Resources:** Yi Zhao, Yuanfa Feng, Anqi Xie, Jun Huang.

**Software:** Lu Zhou, Yiqiang Zhu, lengshan Mo, Jie Lin.

**Supervision:** Jun Huang, Quan Yang.

**Validation:** lengshan Mo, Haixia Wei, Huaina Qiu.

**Visualization:** Lu Zhou, Yiqiang Zhu, Haixia Wei, Quan Yang.

**Writing – original draft:** Lu Zhou, Quan Yang.

**Writing – review & editing:** Jun Huang, Quan Yang.

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
