## [Decision Letter · Decision Letter 0]

26 Jul 2022

Dear Dr. Yang,

Thank you very much for submitting your manuscript "TLR7 controls myeloid-derived suppressor cells expansion and function in the lung of C57BL6 mice infected with Schistosoma japonicum" for consideration at PLOS Neglected Tropical Diseases. Your manuscript was reviewed by members of the editorial board and by several independent reviewers. Thdes have pointed to various issues that require you attention to improve the manuscript substantially. In light of the reviews (below this email), we therefore invite the resubmission of a significantly-revised version that takes into account the reviewers' comments. 

We cannot make any decision about publication until we have seen the revised manuscript and your response to the reviewers' comments. Your revised manuscript is also likely to be sent to reviewers for further evaluation.

Sincerely,

John Pius Dalton, PhD

Academic Editor

Sergio Oliveira

Section Editor

Reviewer's Responses to Questions

**Key Review Criteria Required for Acceptance?**

**Methods**

-Are the objectives of the study clearly articulated with a clear testable hypothesis stated?

-Is the study design appropriate to address the stated objectives?

-Is the population clearly described and appropriate for the hypothesis being tested?

-Is the sample size sufficient to ensure adequate power to address the hypothesis being tested?

-Were correct statistical analysis used to support conclusions?

-Are there concerns about ethical or regulatory requirements being met?

Reviewer #1: The objectives are clearly defined, and the design is appropriate, however, more details should be given in Material and methods, it is not given even the number of mice used in the study. Some methods, ie how the area of granuloma was calculated (how it was measured, number of granuloma measured in each mouse…) is not given in material and methods but it is mentioned in results and in figure 6E. Some details should also be added to the statistical analyses, ie. if the data were parametric or not. This is important to know the appropriate test to apply. The ethical issues are satisfactorily addressed.

Reviewer #2: 1. In Reagents and antibodies, please specify the source of reagents (company, product code).

2. In Parasites and infection, “C57BL/6 mice were infected percutaneously with 40 ± 5 cercariae”. In figure1, “lymphocytes were isolated from S. japonicum infected mouse lung. FACS was used to sort MDSCs, which were then co-cultured with ConA pre-stimulated, CFSE-stained T cells from BALB/c mice at a 1:2 ratio”, Why are MDSCs derived from C57BL/6 mice and T cells from BALB/c mice?

**Results**

-Does the analysis presented match the analysis plan?

-Are the results clearly and completely presented?

-Are the figures (Tables, Images) of sufficient quality for clarity?

Reviewer #1: The results match with the study design and objectives and they are clearly presented, but there are some concerns about the terminology used in histological pictures. Thus, in figure 1B thus authors state that downward arrow indicates interstitial dilation, however this dilation is rounded and similar to that observed in the picture of the control group. These dilations are consistent with central lobular veins, an interstitial dilation due to a lesion usually is irregular in shape, in any case higher resolution should be required for an appropriate identification of these structures. I recommend to delete “and the downward arrow indicates interstitial dilation”. 

In Figure 6D the authors state that upward arrow indicates interstitial dilation, this arrow points an area of interalveolar septal thickening, thus the appropriate terminology here is “interstitial thickening”. In addition, the authors state that the right arrow indicates alveolar thickening while this arrow indicates “alveolar dilation”. 

Since granulomas are typical lesions in schistosomiasis, even the author presents the area of granulomas in Fig. 6E, it would be more interesting for readers a figure showing lung granulomas in both groups instead of a picture showing alveolar dilation and interstitial thickening. 

Otherwise the remaining tables and figures are of good quality.

Reviewer #2: (No Response)

**Conclusions**

-Are the conclusions supported by the data presented?

-Are the limitations of analysis clearly described?

-Do the authors discuss how these data can be helpful to advance our understanding of the topic under study?

-Is public health relevance addressed?

Reviewer #1: Discussion is appropriate and includes recent references on the field of the research. However, conclusions are very brief and the authors have not included conclusions for some of their relevant results, for example the role of TLR7 in pulmonary lesions caused by S. japjonicum.

Reviewer #2: (No Response)

**Editorial and Data Presentation Modifications?**

Reviewer #1: The manuscript should be revised for typographical errors, some of them are:

-line 64: add space after schistosomiasis

-line 72: add space after animals

-line 380: delete space after (Fig 3B)

-line 407: S. japonicum should be in italic

-line 435: delete space after (Fig 6D and 6E)

-line 536: delete space after Md

-line 575: delete space after Immunotherapy

-line 621: delete space after shown

-line 665: delete comma after of and delete space before and after /

Reviewer #2: (No Response)

**Summary and General Comments**

Reviewer #1: The manuscript describes the characteristics of MDSCs in the lung of S. japonicum infected C57BL/6 mice, and the role of TLR7 on the progression of MDSCs activation and differentiation in the lung of S. japonicum infected mice. The work is original and presents new and interesting information to better understand the pathogenesis of schistosomiasis. In general, the paper is well designed and well written, but some more details should be given in material and methods.

Reviewer #2: The manuscript entitled “TLR7 controls myeloid-derived suppressor cells expansion and function in the lung of C57BL6 mice infected with Schistosoma japonicum” by Lu Zhou et al. This study illustrated that TLR7 could delay the progression of S. japonicum infection-induced lung disease mainly through MDSCs. TLR7 deficiency aggravates S. japonicum infection-induced damage in the lung, with more inflammatory cells infiltration, interstitial dilatation and granuloma in the tissue. However, some results or some description are confusing. 

1.Why did the authors choose the sixth week of schistosome infection to observe MDSCs in lung? What are the potential implications?

2.Why did the author investigate TLR7? TLR7 is an endosomal TLR that recognizes single-stranded RNA (ssRNA) and mediates early innate immune responses to viruses, bacteria and malaria. Recently, TLR7 agonists were found to be therapeutics against viral infections and bacteria. SEA and R848 were added to the cells alone or together, and a negative control was used as described in Materials and Methods. Why add soluble egg antigen instead of ssRNA to culture system? 

3.TLR7 deficiency can significantly increase the percentage of PD-L1 or PD-L2- expressed MDSCs in mouse lung after infection (Fig 4B), which indicated that TLR7 deficiency promoted the immunosuppressive function of S. japonicum infection- induced MDSCs by up-regulating the expression of PD-L1 or PD-L2 in mouse lung after infection. However, the results indicated that the lungs of infected TLR7-KO mice showed more infiltration of inflammatory cells, interstitial dilation and granuloma compared with infected WT mice (Fig 6D and 6E), which suggested that TLR7 deficiency might aggravates lung damage in the course of S. japonicum infection. The results seem contradictory.

4.Many descriptions in the background and discussion are unclear, eg, “TLR7 could delay the progression of S. japonicum infection-induced lung disease mainly through MDSCs were involved in the TLR7-mediated immune response”.

PLOS authors have the option to publish the peer review history of their article (what does this mean?). If published, this will include your full peer review and any attached files.

Reviewer #1: No

Reviewer #2: No
---

## [Decision Letter · Decision Letter 1]

27 Sep 2022

Dear Dr. Qang Yang,

We are pleased to inform you that your manuscript 'TLR7 controls myeloid-derived suppressor cells expansion and function in the lung of C57BL6 mice infected with Schistosoma japonicum' has been provisionally accepted for publication in PLOS Neglected Tropical Diseases.

Best regards,

John Pius Dalton, PhD

Academic Editor

Sergio Oliveira

Section Editor

Reviewer's Responses to Questions

**Key Review Criteria Required for Acceptance?**

**Methods**

-Are the objectives of the study clearly articulated with a clear testable hypothesis stated?

-Is the study design appropriate to address the stated objectives?

-Is the population clearly described and appropriate for the hypothesis being tested?

-Is the sample size sufficient to ensure adequate power to address the hypothesis being tested?

-Were correct statistical analysis used to support conclusions?

-Are there concerns about ethical or regulatory requirements being met?

Reviewer #2: (No Response)

**Results**

-Does the analysis presented match the analysis plan?

-Are the results clearly and completely presented?

-Are the figures (Tables, Images) of sufficient quality for clarity?

Reviewer #2: (No Response)

**Conclusions**

-Are the conclusions supported by the data presented?

-Are the limitations of analysis clearly described?

-Do the authors discuss how these data can be helpful to advance our understanding of the topic under study?

-Is public health relevance addressed?

Reviewer #2: The conclusions are reasonable.

**Editorial and Data Presentation Modifications?**

Reviewer #2: (No Response)

**Summary and General Comments**

Reviewer #2: (No Response)

PLOS authors have the option to publish the peer review history of their article (what does this mean?). If published, this will include your full peer review and any attached files.

Reviewer #2: No

---

## [Editor Report · Acceptance letter]

7 Oct 2022

Dear Associate professor Yang,

We are delighted to inform you that your manuscript, "TLR7 controls myeloid-derived suppressor cells expansion and function in the lung of C57BL6 mice infected with *Schistosoma japonicum*," has been formally accepted for publication in PLOS Neglected Tropical Diseases.

Best regards,

Shaden Kamhawi

co-Editor-in-Chief

Paul Brindley

co-Editor-in-Chief
